# Efficacy of Prophylactic Antibiotics in COPD: A Systematic Review

**DOI:** 10.3390/antibiotics13121110

**Published:** 2024-11-21

**Authors:** Anh Tuan Tran, Amr Sayed Ghanem, Marianna Móré, Attila Csaba Nagy, Ágnes Tóth

**Affiliations:** 1Department of Integrative Health Sciences, Institute of Health Sciences, Faculty of Health Sciences, University of Debrecen, 4028 Debrecen, Hungary; anhtran310795@gmail.com; 2Department of Health Informatics, Institute of Health Sciences, Faculty of Health Sciences, University of Debrecen, 4028 Debrecen, Hungary; aghanem@etk.unideb.hu (A.S.G.); attilanagy@med.unideb.hu (A.C.N.); 3Department of Gerontology, Institute of Social and Sociological Sciences, Faculty of Health Sciences, University of Debrecen, 4400 Nyíregyháza, Hungary; more.mariann@etk.unideb.hu

**Keywords:** chronic obstructive pulmonary disease, COPD, prophylaxis, antibiotics

## Abstract

**Background/Objectives**: Chronic obstructive pulmonary disease (COPD) is a global health problem and the third leading contributor to mortality worldwide. This systematic review aims to summarize the results of previous studies tackling the question of the efficacy of long-term prophylaxis of antibiotics in COPD patients, with particular regard to exacerbation rate, time to first exacerbation, health status, airway bacterial load, inflammatory markers, cell counts in sputum samples, and potential adverse events. **Results**: Four studies found significant improvement in the exacerbation rate in patients receiving antibiotic intervention. One study found doxycycline to have negative effects on patients’ exacerbation outcomes. Two studies recorded a reduction in total airway bacterial load using quantitative culture of sputum samples, but the prevalence of antibiotic-resistant bacteria increased in all studies that measured it. No change in inflammatory markers was observed; however, there was a decline in neutrophil cell counts and, subsequently, reductions in neutrophil elastase concentrations. **Methods**: PubMed and Web of Science databases were searched for English-language studies presenting data on the prophylactic use of antibiotics in COPD management. All included studies are randomized controlled trials (RCTs) and meet the inclusion criteria. **Conclusions**: Based on current evidence from RCTs, the prophylactic antibiotic approach utilizing macrolides is the most effective in reducing the incidence of COPD exacerbation. However, the emergence of antibiotic-resistant pathogens is notable. Whether the beneficial effects of macrolides on exacerbation are due to their antibacterial or immunomodulant properties is still inconclusive. Future studies are needed to better understand the interactions between antibiotics and the airway microbiome during COPD exacerbation.

## 1. Introduction

Chronic obstructive pulmonary disease (COPD) is a progressive, heterogeneous respiratory disease distinguished by long-lasting respiratory symptoms, which include shortness of breath, persistent cough, sputum production, and/or occasional exacerbations. These symptoms result from abnormalities in the airways or structural changes accompanied by chronic inflammation. The condition is irreversible, and these abnormalities lead to a sustained and frequent worsening of airflow blockage [1].

COPD is the third leading cause of death worldwide. It was responsible for 392 million cases of COPD globally, associated with 3.23 million deaths and 74.4 million disability-adjusted life years (DALYs) in 2019 [2]. Most prevalence studies use the Global Initiative for Chronic Obstructive Lung Disease (GOLD) criteria for COPD, with a post-bronchodilator ratio of FEV_1_/FVC < 0.7, where FEV_1_ is the forced expiratory volume during the first second after maximum inspiration, and FVC is the forced vital capacity [1]. This criterion has a possible drawback of overdiagnosis in the elderly population due to the ratio decreasing with age. The prevalence increases particularly in developing countries due to increases in smoking rates and a decline in other causes of death [3]. Studies have suggested that there is a greater mortality rate in men compared to women; however, the trend of mortality has either increased at a lower rate or decreased in men compared to women in the studied countries [4].

COPD can be a result of a wide variety of causes, ranging from genetics [5], abnormal lung development, and infections to environmental pollutants [6], or even a mix of several factors [7]. The greatest contributor to the development of COPD is cigarette smoking [1]. Yet fewer than 50% of heavy smokers develop COPD, so other factors, including biomass exposures, air pollution, and occupational exposures, need to be considered [1,8,9]. Continuous exposure to irritants such as tobacco smoke leads to chronic inflammation that can be frequently observed in COPD patients.

The inflammatory response in patients with COPD has a characteristic pattern of an elevated number of activated neutrophils, macrophages, and lymphocytes compared to normal levels. This is due to inhaled particulates triggering a response from surface macrophages and epithelial cells of the respiratory tract to release chemotactic mediators for the recruitment of circulating neutrophils and lymphocytes to the lung tissue. The release of pro-inflammatory cytokines, e.g., IL-6, IL-8 (CXCL8), and IL-10, further amplifies the process [10]. With the increase in the recruitment of neutrophils to the lung tissue in response to chronic inflammation, it is suggested that this also leads to the imbalance between the protease activity of neutrophil elastase and alpha-1 antitrypsin that counteracts it [11]. The uncontrolled activity of neutrophil elastase may result in the degradation of lung interstitial elastin. Once destroyed, the development of emphysema and loss of lung elasticity will occur. The persistent inflammatory response also causes structural changes, such as a reduction in airway lumen due to increased airway wall thickness and mucus production. This is coupled with an extensive loss of alveolar gas exchange surface and pulmonary capillaries due to emphysema, resulting in abnormal ventilation–perfusion (V_A_/Q) distribution. Alveolar ventilation and pulmonary blood flow mismatches are the main mechanism for arterial hypoxemia, with or without hypercapnia, in various stages of COPD [12]. Due to the inflammation, C-reactive protein (CRP) levels are elevated in COPD patients. The severity of COPD is directly related to elevated CRP levels, which can contribute to identifying patients [13].

Chronic dyspnea is the most characteristic symptom and a primary source of impairment associated with the disease. Dyspnea is a complex symptom in COPD patients, with many factors potentially involved in its pathogenesis, including impaired respiratory effort due to airflow obstructions, gas exchange abnormalities, peripheral muscle dysfunction, psychological distress, and cardiovascular or other comorbidities. In the majority of cases, chronic cough is the first symptom to appear in COPD. Patients with COPD may or may not produce sputum while coughing. Consistent production of sputum is an indication of increased inflammatory response and may represent the onset of bacterial exacerbation. The symptoms of wheezes, chest tightness, and fatigue are irregular in nature, and patients may experience these symptoms spontaneously during the week and over the course of a day. Chest tightness often arises following patients’ attempts at physical activity and may result from isometric contraction of intercostal muscles. Fatigue, a feeling of tiredness and exhaustion, is one of the most common symptoms of COPD. It limits the patients’ daily function and heavily impacts their quality of life [1].

Exacerbations in COPD (ECOPD) are short periods of increased coughing, dyspnea, and increased sputum production that sometimes can be purulent. It is often accompanied by enhanced local and systemic inflammation caused by inhaled irritants or, more prevalently, viral and/or bacterial infections. It is estimated that 60–80% of all exacerbations involve infections from bacteria, viruses, or both [3]. The lower airway of the majority of COPD patients is colonized with various species of bacteria. Colonizing bacteria in the airway promote bronchial inflammation, leading to respiratory epithelial injury, which, in turn, increases the risk of acquiring new microbial strains and exacerbation [3]. The frequency of ECOPD varies from patient to patient, but it may be higher in more severe cases of COPD [14]. It was shown that rhinovirus/enterovirus and influenzavirus are the most common root causes of acute exacerbations in COPD (AECOPD) patients [15]. Viral infections account for 48.4% of infectious exacerbations. It was found that infectious exacerbations, especially those involving coinfection of bacteria and viruses, leave a greater negative impact on patients’ lung function and cause prolonged hospitalization compared to non-infectious exacerbations [16]. Therefore, influenza vaccination is recommended in COPD guidelines [17].

There is currently no cure for COPD. Since exposure to tobacco smoke is the top contributor leading to the onset of COPD, smoking cessation is the primary intervention that will have the most significant impact on the progression of the disease. Treatment options for COPD include pharmacological and non-pharmacological therapies with the main aim of alleviating symptoms, minimizing the frequency and severity of exacerbations, and improving patients’ health status. Non-pharmacological treatments range from pulmonary rehabilitation, oxygen therapy [1], and ventilation support [18,19] to surgical interventions such as lung volume reduction [20] or lung transplantation [21]. Pharmacological treatments include a wide variety of medications such as bronchodilators and antimuscarinic drugs. Two major types of bronchodilators are beta_2_-adrenergic agonists and anticholinergics. Both beta_2_-agonists and antimuscarinics are available in short-acting forms (SABAs and SAMAs) and long-acting forms (LABAs and LAMAs). Furthermore, inhaled corticosteroids, phosphodiesterase-4 inhibitors, and antibiotics could be involved in the management of the disease if infection is confirmed [1,3,22].

Prophylactic antibiotic treatment is still not recommended in COPD patients with frequent exacerbations [1]. One of the greatest concerns is that long-term antibiotic use may increase the risk of developing bacterial resistance [23]. In addition, it may reduce the microbial diversity of the respiratory tract as well [24,25]. This systematic review aims to evaluate the effects of antibiotic prophylaxis on COPD patients’ exacerbations and quality of life, as well as safety concerns related to long-term administration of antibiotics. Our specific research questions were as follows: whether long-term antibiotic use leads to changes in (1) exacerbation rate, (2) time to first exacerbation, (3) health status, (4) total airway bacterial load, (5) inflammatory markers and neutrophil cell counts in sputum samples of patients, and (6) whether a substantial difference in the overall number of adverse events is detectable in antibiotic prophylaxis compared to placebo.

## 2. Results

The conducted searches yielded 316 records from PubMed and 256 records from Web of Science. A total of 87 records were removed due to duplication, and 2 records were removed due to reports being retracted. A total of 483 records were screened for titles and abstracts, of which 451 records were excluded. A total of 32 full-text reports were evaluated for eligibility; 3 reports could not be retrieved. A total of 13 reports from studies met the inclusion criteria and were included in the systematic review. A total of 11 reports were excluded due to incorrect study design (e.g., not primary research, re-evaluation, or retrospective observational studies), 4 reports were excluded due to incorrect study population (patients with comorbidities, e.g., asthma, bronchiectasis, cystic fibrosis, antibody deficiency, tracheostomy, or patients under the age of 18), and 1 report was excluded for not being in English (Figure 1, Table 1 and Table 2).

### 2.1. Overview of Included Studies

Thirteen studies were eligible for the systematic review. Eight studies [26,27,28,31,33,35,36,37] used continuous prophylaxis, where antibiotics were administered on a daily basis. Three studies [29,32,38] administered antibiotics three times per week. One study [34] utilized intermittent antibiotic prophylaxis, where patients were given antibiotics five days per week, repeated every eight weeks for a total of six courses. Another study [30] applied continuous prophylaxis and administered antibiotics three times a week while also utilizing intermittent prophylaxis.

The antibiotics investigated were azithromycin [26,29,31,36,38], erythromycin [32,33,37], doxycycline [27], clarithromycin [28], moxifloxacin [34], or multiple antibiotic classes [30,35].

Eleven studies [26,27,28,29,31,32,33,34,35,36,38] listed exacerbation rate as a primary or secondary outcome. Nine studies [26,27,28,29,30,31,32,35,38] evaluated patients’ health status in response to antibiotic prophylaxis. Six studies [30,32,33,36,37,38] examined inflammatory markers and cell counts. Five studies measured bacterial load in sputum samples [26,28,30,34,36].

### 2.2. Study Participants

All the included studies used the same definition of clinical diagnosis of COPD as having a post-bronchodilator ratio of FEV_1_/FVC < 0.7 or FEV_1_% predicted < 80%, with a smoking history of at least 10 packs per year. Since the physiological changes associated with exacerbation were found to return to baseline in about 4 weeks in the majority of patients [39], the study participants were also required to undergo a run-in period of 4 weeks without any reported exacerbation or alteration in respiratory medication to ensure stability before being involved in the study. Exclusion criteria were consistent across studies and excluded patients with pulmonary comorbidities, patients with cardiovascular disease, and patients allergic to/or taking medication that could adversely interact with the trial antibiotics. The studies also noted that patients were required to continue with their usual medications, including bronchodilators and inhaled corticosteroids. Any change in the intervention was due to clinical necessity, in which case the patient was excluded from the study.

Recruited participants in the study by Banerjee et al. [28] had to be in moderate to severe COPD classification, while Tan et al. [37] required participants to be in GOLD stages II-IV. Simpson et al. [36] required participants to have COPD with persistent neutrophilic bronchitis, which was defined as “sputum neutrophil proportion of more than 61% or more than 162 × 10^4^/mL sputum neutrophil demonstrated on two occasions (at least one being the screening visit)” due to the objective of the study. In order to recruit more participants likely to experience AECOPD, Han et al. [31] required participants to either be under continuous oxygen supplementation or to have had an AECOPD within the previous 12 months before participating in the study.

The total number of participants was 2309 in the antibiotic intervention groups and 2155 in the placebo groups recruited from countries including the United States of America, the United Kingdom, the Netherlands, China, Australia, and New Zealand.

### 2.3. Exacerbation

Out of the included studies, four studies found that there was a significant reduction in the rate of exacerbation during the duration of the trials: Albert et al. [26] reported an acute exacerbation rate in the treatment arm of 1.48 per patient-year and in the placebo arm of 1.83 per patient-year, giving a rate ratio of 0.83 (95% CI, 0.72 to 0.95; *p* = 0.01). He et al. [32] reported the relative risk in the antibiotic group compared to the placebo group as 0.554 (95% CI, 0.314 to 0.979; *p* = 0.042). Seemungal et al. [33] showed that the rate ratio of moderate to severe exacerbation compared to the placebo was 0.648 (95% CI, 0.489 to 0.859; *p* = 0.003). It was also noted that the observed exacerbations were more frequent in patients with lower FEV_1_% predicted at enrollment or patients with a history of frequent exacerbations. Uzun et al. [38] observed a significant reduction in exacerbation rate in the azithromycin group compared to the placebo group with an adjusted rate ratio of 0.58 (95% CI, 0.42 to 0.79; *p* = 0.001).

Three studies found lower exacerbation rates; however, these reductions were not significant: Allinson et al. [27] concluded that there was no significant difference between doxycycline and placebo groups, with an adjusted rate ratio of 0.85 (95%CI, 0.67 to 1.07; *p* = 0.16). Simpson et al. [36] reported a rate ratio of 0.38 (95% CI, 0.14 to 1.05; *p* = 0.062) compared to the placebo group. Brill et al. [30] reported a rate ratio of 0.72 (0.30 to 1.71; *p* = 0.45) in the azithromycin arm. However, Brill et al. [30] also reported an increase in exacerbation in the patient group with doxycycline compared to the placebo group, with an adjusted relative risk of 2.07 (95% CI, 0.99 to 4.35; *p* = 0.05) (Table 3).

### 2.4. Time to First Exacerbation

Studies also showed that compared to placebo, groups receiving antibiotics also had statistically significant increases in time to first exacerbation. Seemungal et al. [33] showed that the median time to first exacerbation in the erythromycin group was 271 days versus 89 days in the placebo group (*p* = 0.02; log-rank test). He et al. [32] also reported similar results, with the median time to first exacerbation in the erythromycin group being 155 days compared to 86 days in the placebo group (*p* = 0.032; log-rank test). Albert et al. [26] reported a median time to first exacerbation of 266 days (95% CI, 277 to 313) in the azithromycin arm compared to 174 days (95% CI, 143 to 215) in the placebo arm (*p* < 0.001). Similarly, Uzun et al. [38] reported the median time to first exacerbation as 59 days (95% CI, 31 to 87) in the placebo group and 130 days (95% CI, 28–232) in the azithromycin group (*p* = 0.001).

### 2.5. Health Status

The majority of included studies used SGRQ and/or SF-36 to assess the effect of the intervention on the health status outcome of the participants. SGRQ is a self-administered questionnaire developed to measure health status (quality of life) in patients with COPD or asthma. It is also used in settings such as randomized clinical trials or population surveys. The questionnaire comprises 50 items that survey the patients’ recollection of their symptoms (symptom score), disruption in their physical activity (activity score), and psychological impact (impact score). The total score ranges from 0 to 100, with lower scores indicating better functioning. A minimum change of 4 units in the score is required to be considered clinically relevant.

Of the included studies, only two studies reported clinically relevant improvements in SGRQ total score at the end of interventions. Berkhof et al. [29] reported a mean difference of −7.4 units (95% CI, −12.5 to −2.5; *p* = 0.004) compared to placebo. Uzun et al. [38] also reported a similar effect, with the azithromycin group’s SGRQ total score mean difference compared to placebo at 3 months being −4.2 units (95% CI, −8.3 to −0.1; *p* = 0.043); however, the author noted that the change did not last for 12 months. In the study conducted by Brill et al. [30], there was an improvement in the SGRQ total score in the moxifloxacin and azithromycin groups, as well as a decline in the doxycycline group. However, these changes were not statistically significant, nor did they reach the minimum change of 4 units to be clinically relevant. In contrast to the previous study, Allinson et al. [27] measured a worse SGRQ total score mean difference for participants taking doxycycline (an increase of 5.2 units (95% CI, 1.44 to 9.00; *p* = 0.007)) compared to those taking placebo (Table 4).

Although no significant improvement was measured in SGRQ total scores, some studies did observe a significant improvement in SGRQ component scores. Banerjee et al. [28] reported an improvement in symptom scores, with a mean difference between the clarithromycin arm and the placebo arm of −10.2 units (95% CI, 1.6 to 18.7; *p* = 0.04). Berkhof et al. [29] also reported similar findings, with mean differences between azithromycin and placebo groups in component symptom score and impact score being −9.1 units (95% CI, −17.6 to −0.7; *p* = 0.034) and −8.9 units (95% CI, −14.5 to −3.3; *p* = 0.002), respectively. Uzun et al. [38] recorded a significant difference in symptom scores between the azithromycin group and the placebo group of −5.06 units (955 CI, −9.64 to −0.49; *p* = 0.03). Albert et al. [26] also recorded a clinically relevant reduction in SGRQ score (43% vs. 36%; *p* = 0.03) in the azithromycin group compared to placebo.

### 2.6. Bacteriology

Two studies conducted bacterial load analysis using the quantitative culture method of sputum samples. Both utilized a modified version of the method described by Pye et al. [40] in their analysis. Brill et al. [30] reported an adjusted mean change in total bacterial load compared to placebo in all moxifloxacin, doxycycline, and azithromycin arms, with values of −0.42 log_10_ cfu/mL (95% CI, −0.91 to 0.08; *p* = 0.10), −0.11 log_10_ cfu/mL (95% CI, −0.55 to 0.33; *p* = 0.62), and −0.08 log_10_ cfu/mL (95% CI, −0.54 to 0.39; *p* = 0.73), respectively. However, none of these changes were statistically significant. In the study conducted by Simpson et al. [36], median total bacterial load reduction occurred in both the intervention and placebo arms, with median bacterial load reductions of 53% (from 5.02 cfu/mL to 2.37 cfu/mL) in the azithromycin group and 37% (from 6.55 cfu/mL to 4.14 cfu/mL) in the placebo group.

Albert et al. [26] focused on a selected number of respiratory pathogens, including *Staphylococcus aureus*, *Streptococcus pneumoniae*, *Haemophilus* spp., and *Moraxella* spp. Albert et al. [26] reported that 12% of participants in the azithromycin group and 31% of participants in the placebo group who had not had nasopharyngeal colonization at the time of enrollment became colonized during the course of the study (*p* < 0.001), but no association was reported between nasopharyngeal colonization and the occurrence of exacerbation.

In studies that assessed antibiotic resistance of colonizing bacteria, mean inhibitory concentrations (MICs) were established for each isolate against the antibiotics in question. Isolates were classified as sensitive, intermediate, or resistant if breakpoints were reached. In a study conducted by Brill et al. [30], isolated cultures from both pre- and post-administration were analyzed for antibiotic resistance. Adjusting for baseline factors and whether species could be associated with lower respiratory tract infections, factor increases in MIC of 4.82 (95% CI, 1.44 to 16.19; *p* = 0.01) for moxifloxacin, 3.74 (95% CI, 1.46 to 9.58; *p* = 0.01) for doxycycline, and 6.23 (95% CI, 1.66 to 23.35; *p* = 0.01) for azithromycin were reported compared to placebo [30]. It was also noted that, compared to placebo, patients treated with doxycycline were more likely to develop resistance to doxycycline, with an OR of 5.77 (95% CI, 1.40 to 23.74; *p* = 0.02) [30]. The ORs were 2.03 (95% CI, 0.36 to 11.54; *p* = 0.42) and 2.42 (95% CI, 0.61 to 9.62; *p* = 0.21) for the moxifloxacin and azithromycin groups, respectively, although these were not statistically significant [30]. Of those who had not had nasopharyngeal colonization at the time of enrollment but had bacterial colonization during the course of the study, Albert et al. [26] found that the incidence of macrolide resistance in selected pathogens was 81% in the azithromycin group and 41% in the placebo group (*p* < 0.001).

### 2.7. Inflammatory Markers and Cell Count

In the study conducted by Brill et al. [30], sputum inflammatory markers IL-6 and IL-8 were measured in the intervention and placebo arms. For IL-6, the differences in estimated means were −0.16 log_10_ pg/mL (95% CI, −0.50 to 0.18; *p* = 0.37), −0.21 log_10_ pg/mL (95% CI, −0.56 to 0.13; *p* = 0.22), and −0.19 log_10_ pg/mL (95% CI, −0.43 to 0.27; *p* = 0.65) in the moxifloxacin, doxycycline and azithromycin groups, respectively, compared to placebo [30]. For IL-8, the differences in estimated means were −0.26 log_10_ pg/mL (95% CI, −0.72 to 0.19; *p* = 0.26), −0.11 log_10_ pg/mL (95% CI, −0.55 to 0.34; *p* = 0.64), and 0.00 log_10_ pg/mL (95% CI, −0.46 to 0.46; *p* = 1.00) [30] in the moxifloxacin, doxycycline and azithromycin groups, respectively, compared to placebo. A significant change was not observed in the cytokines IL-6 and IL-8 in 3 in any of the three antibiotic arms compared to placebo. In the study by Seemungal et al. [33], the estimated means of sputum inflammatory markers IL-6 and IL-8 were measured at the first visit for baseline and at 12 months. The estimated mean of IL-6 changed from 174 pg/mL (95% CI, 120 to 228) to 128 pg/mL (95% CI 63 to 194; *p* = 0.514) after 12 months of erythromycin administration [33]. The estimated mean of IL-8 changed from 3095 pg/mL (95% CI, 2678 to 3514) to 3138 pg/mL (95% CI, 2576 to 3699; *p* = 0.551) after 12 months of intervention [33]. Again, no significant changes were observed in sputum inflammatory markers.

He et al. [32] reported a significant reduction in total cell counts from 3.74 × 10^6^/mL (95% CI, 3.24 to 4.24) at baseline to 3.19 × 10^6^/mL (95% CI, 2.84 to 3.53; *p* = 0.005) after 3 months with erythromycin, and to 3.20 × 10^6^/mL (95% CI, 2.81 to 3.59; *p* = 0.004) after 6 months. Neutrophil cell counts also exhibited a similar significant reduction from baseline at 2.80 × 10^6^ (95% CI, 2.40 to 3.20) to 2.29 × 10^6^ (9% CI, 1.87 to 2.70; *p* = 0.002) at 3 months and to 2.25 × 10^6^ (95% CI, 1.90 to 2.60; *p* = 0.001). Furthermore, a reduction in neutrophil elastase concentration in sputum supernatant compared to baseline was also observed in this study (*p* = 0.027 after 3 months and *p* = 0.015 after 6 months) [32].

### 2.8. Adverse Events

Overall, no study reported substantial differences in the overall number of adverse events. The most frequent ones were gastrointestinal disorders such as diarrhea [27,29,33,34,36,38], nausea [27,33,34,35,38], vomiting [27,33,34,38], abdominal pain [33,34,37], dyspepsia [29,33], and alteration of taste [29]. Albert et al. [26] reported that hearing decrement occurred in 142 patients in the azithromycin group (25%) and 110 patients in the placebo group (20%) (*p* = 0.04), confirmed by audiogram. Hearing decrement was also detected in 25% of patients receiving azithromycin versus 20% of those receiving placebo (*p* = 0.04) by Han et al. [31].

In a few cases, urticaria [33,34], hypersensitivity, abnormal electrocardiograms [35], and one case of left-sided heart failure [37] were detected.

In the study by Albert et al. [26], the rate of death from any cause was 3% in the azithromycin group and 4% in the placebo group, while in the study by Sethi et al. [34], the mortality rate was 0.3% (1/351) in the moxifloxacin group and 0.8% (3/387) in the placebo group.

## 3. Discussion

Most studies conducted with the macrolide class of antibiotics reported similar results regarding exacerbation rates, although there was an increase in exacerbation rate in the Brill et al. [30] study. The authors noted that the number of patients in all arms of the investigation was smaller than in other studies, the study only lasted 13 weeks, and the study design was not focused on the exacerbation rate as the primary outcome, all of which could impact the ability to accurately determine the effect of antibiotics on the exacerbation rate.

The median times to first exacerbation in antibiotic groups in studies conducted by Albert et al. [26] and Seemungal et al. [33] were similar, at 266 days and 271 days, respectively. However, the placebo group in the study by Albert et al. [26] had a longer median time to first exacerbation than in the study by Seemungal et al. [33] (174 days compared to 89 days). This may be because Seemungal et al. [33] recruited patients with moderate to severe COPD, with 38 (34%) patients experiencing three or more episodes of exacerbations in the year prior to enrollment. In the study by He et al. [32], the median time to first exacerbation was 155 days with erythromycin intervention, which was shorter than the 266 days reported by Seemungal et al. [33] with the same antibiotic. The explanation for this may be the higher proportion of patients taking LAMA or LABA in the study by He et al. [32], and the shorter study duration, which could have affected the outcome of the measurement.

Based on the results, azithromycin and erythromycin appeared to be very effective in reducing exacerbation rates and delaying the time to first exacerbation; meanwhile, roxithromycin, doxycycline, and moxifloxacin failed to produce any statistical difference in exacerbation rates when compared to placebo. In addition, doxycycline and moxifloxacin even showed an increase in exacerbation rates in the study by Brill et al. [30]. This suggests that the tetracycline and fluoroquinolone classes of antibiotics are ineffective for long-term or pulse treatment of exacerbation, at least in regard to reducing the exacerbation rate. Meanwhile, the macrolide class of antibiotics exhibited effective results in exacerbation rate reduction and increased the time to first exacerbation. However, it should be noted that not all macrolide antibiotics have the same effectiveness. In studies performed by Albert et al. [26] and Uzun et al. [38], azithromycin interventions were found to be significantly effective in exacerbation rate reduction, similar to studies performed by He et al. [32] and Seemungal et al. [33] in which erythromycin was compared to placebo. Although Shafuddin et al. [35] also showed reductions in exacerbations in either the roxithromycin group alone or in conjunction with doxycycline, the decreases were statistically insignificant.

In regard to health status, doxycycline was reported to worsen the quality of life measured by the SGRQ score in contrast to improvements in those receiving a placebo. The reasons behind the lack of success of doxycycline remained unclear in the study; however, the authors also noted that the difference may be explained by the withdrawal of sicker patients taking the placebo [27]. Studies that reported significant improvements in SGRQ total scores are those that compared azithromycin and placebo. These studies also reported noticeable improvements in symptom scores and impact scores [29,38]. In the study comparing clarithromycin to placebo, although there was no clinically relevant reduction in total SGRQ scores, there was still a significant improvement in symptom scores compared to placebo [28]. This effect on symptom or impact scores but not total scores may indicate that the macrolide class of antibiotics is effective in reducing the rate of exacerbations, thus improving symptom scores compared to other classes of antibiotics.

In the study that measured the total bacterial load by quantitative sputum cultures, a reduction in airway bacteria was recorded in all antibiotic arms. In the study by Brill et al. [30], although there was a reduction in total airway bacterial load, no reduction in exacerbation rates was reported, and there was even an increase in exacerbation rates in the group receiving doxycycline. In contrast, the study conducted by Simpson et al. [36] reported both a reduction in airway bacterial numbers and exacerbation rates. Albert et al. [26] focused on bacteria associated with exacerbation instead and found that participants in the azithromycin group acquired fewer nasopharyngeal colonizations compared to the placebo during the course of the study. This was also accompanied by a significant reduction in exacerbation rates in the intervention group compared to placebo. Although long-term administration of antibiotics does reduce total bacterial load in the airway, this change does not accompany any change in exacerbation rates. The exacerbation rate reduction observed in other studies may not be caused by changes in total bacterial load but rather by the change in the quantity of bacteria that is often associated with COPD exacerbation.

Patients in antibiotic intervention groups were more likely to be colonized by antibiotic-resistant pathogens. The underlying mechanisms in macrolide resistance may include post-transcriptional methylation of bacterial 23S ribosomal RNA [41], ribosomal mutations that prevent antibiotic binding, the presence of efflux pumps [42] and ATP-binding cassette F (ABC-F) proteins [43], drug modification, and inhibition by phosphotransferases and esterases [44]. In the case of quinolones and fluoroquinolones, mutations in DNA gyrase (topoisomerase IV) decrease the opportunity for complex formation, and the presence of efflux pumps and proteins that protect DNA gyrase and topoisomerase IV can be major mechanisms contributing to resistance [45]. In contrast, tetracycline resistance is attributed to efflux pumps, ribosomal mutations, protein-mediated ribosome protection, and enzymatic inactivation [46].

There was no significant change in inflammatory marker status in sputum samples of patients receiving antibiotics. In the study conducted by He et al. [32], it was noted that the erythromycin group exhibited a significant reduction in total cell counts, contributed by a decrease in neutrophils. Notably, the reduction in neutrophil cell counts also resulted in a significant reduction in neutrophil elastase concentrations in sputum supernatant [32]. This reduction in neutrophil elastase may induce the beneficial effect of reducing proteolytic damage to the lung interstitial elastin. This effect may also contribute to the reduction in the exacerbation rates seen in this study.

No significant differences in the frequency of adverse events were observed when comparing the treatment group with the placebo group across all studies. The most common complaints were gastrointestinal disorders, nausea, vomiting, diarrhea, and hearing decrements.

The majority of the included studies utilized the macrolide class of antibiotics for long-term prophylaxis of COPD exacerbations due to their antibiotic and immunomodulant effects. There are many mechanisms that were assumed to be behind the strong immunomodulatory effects of macrolides in both innate and adaptive aspects of immunity. These might be attributed to mechanisms such as increasing the production of anti-inflammatory cytokine IL-10 by inducing expression of M2 macrophages while decreasing levels of IL-12 and other pro-inflammatory molecules by inhibition of nuclear factor kappa B (NF-κB), which regulates chemokine and cytokine expression. It is also thought that macrolides promote the phagocytic ability of macrophages while attenuating responses and inducing apoptosis in neutrophils. This may be the possible explanation for the reduction in neutrophil cell counts observed by He et al. [29]. An additional beneficial effect of macrolides in respiratory diseases has been reported as the modulation of NETosis, the process by which NETs (neutrophil extracellular traps)—web-like structures of chromatin, histones, and granular proteins designed to trap bacteria—are released, but they can also cause significant tissue damage. Dendritic cells (DCs) can also cause tissue damage and maintain chronic inflammation in inflammatory diseases. It has been suggested that macrolides may polarize DCs into a tolerogenic phenotype that can also lead to suppression of T-cell activation/proliferation or induction of the formation of anti-inflammatory T_reg_ cells [47].

Azithromycin and clarithromycin are preferred over erythromycin due to their more tolerable side effect profiles. Azithromycin and clarithromycin also exhibit a long half-life and slow release, resulting in lower doses required and better patient adherence. Macrolides are active against respiratory pathogens that are suspected to be associated with COPD exacerbations, such as *Haemophilus influenzae*, *Streptococcus pneumoniae*, *Moraxella catarrhalis*, and *Staphilococcus aureus*. Tetracyclines and fluoroquinolone antibiotics did not demonstrate any beneficial effects in the included studies. Doxycycline could also be a candidate for antibiotic prophylaxis in COPD exacerbations due to its antibacterial and anti-inflammatory properties; however, Brill et al. [30] found that doxycycline not only does not result in any significant reduction in inflammatory cytokines but also increases exacerbation rates. A study performed by Allinson et al. [27] was better equipped to measure exacerbation rates with its larger participant pool and longer duration of treatment: with the same dosage and regimen, it was found that doxycycline reduced exacerbation rates, although not to a statistically significant degree. In addition, patients receiving doxycycline reported a worsening in their health status compared to the placebo group. Unfortunately, the study did not measure any outcome related to inflammation.

## 4. Methodology

### 4.1. Types of Studies

We included studies that were randomized controlled trials (RCTs). The included studies had to be retrievable and reported in full text.

### 4.2. Types of Participants

We included participants from the adult population (older than the age of 18) with a diagnosis of COPD as defined by GOLD, the American Thoracic Society, or the European Respiratory Society (post-bronchodilator FEV_1_/FVC < 0.7 confirmed by spirometry). We excluded participants who presented with co-morbidities such as asthma, bronchiectasis, or cystic fibrosis, which may also lead to airflow restriction.

### 4.3. Types of Interventions

We included studies that compared antibiotic prophylaxis with placebo. The eligible studies had to have randomized participants to receive antibiotics or a placebo. Antibiotic prophylaxis could be continuous or intermittent.

### 4.4. Types of Outcome Measures

#### 4.4.1. Primary Outcomes

Exacerbations. We extracted data on the exacerbation rate, time to first exacerbation, or both, if available.Health status (validated by St. George’s Respiratory Questionnaire).

#### 4.4.2. Secondary Outcomes

Bacteriology;Inflammatory markers;Adverse events.

### 4.5. Search Strategy

We conducted searches on PubMed and Web of Science for English-language studies presenting data related to the prophylactic use of antibiotics in the management of exacerbations in COPD patients. The last search was conducted in February 2024. The respective search strings were applied to searches conducted on PubMed and Web of Science.

PubMed

#1(((((((((“Anti-Bacterial Agents”[Mesh]) OR (antibiotic)) OR (amoxicillin)) OR (levofloxacin)) OR (moxifloxacin)) OR (tetracycline)) OR (doxycycline)) OR (azithromycin)) OR (erythromycin)) OR (clarithromycin).#2((“Pulmonary Disease, Chronic Obstructive”[Mesh]) OR (COPD)) OR (“Chronic Obstructive Pulmonary Disease”).#3(#1) AND (#2) Filters: Randomized Controlled Trial.

Web of Science

Chronic obstructive pulmonary disease (All Fields) and Random* Controlled Trial (All Fields) and antibiotic* (All Fields).

### 4.6. Selection of Studies

Results from all databases were combined into a Zotero library. Titles and abstracts were scanned to identify eligible studies. The complete texts of possible eligible articles were retrieved and screened to ensure conformity with inclusion criteria.

### 4.7. Data Extraction

The following data were extracted from the included studies:Methods: study design, duration of study.Participants: number of participants, age, stage of COPD.Intervention: type of antibiotics, concomitant medication.Outcomes: primary and secondary outcomes as specified.

In statistical analyses of the included studies, *p* values < 0.05 were considered to be significant.

## 5. Conclusions and Future Directions

Based on current evidence from RCTs, the macrolide class of antibiotics appears to be the most effective in reducing exacerbation rates in COPD patients. Although its effect on airway inflammation is inconclusive, its outcome on neutrophil cell counts is very noticeable. No antibiotics resulted in any significant improvement in lung function or exercise capacity at the end of the interventions. Although there was a reduction in airway bacterial load, varying degrees of antibiotic-resistant pathogens were detected in the included studies.

Azithromycin appears to be the most promising because of its better patient adherence, dose management, and more tolerable side effects compared to erythromycin. Furthermore, the risk of drug–drug interactions is lower compared to other macrolides, such as erythromycin and clarithromycin [48]. Additionally, it displays potent immunomodulatory properties [49]. However, the administration of azithromycin should be very cautious; known hypersensitivity, allergies, cardiovascular conditions, or other co-morbidities should be taken into serious consideration. In addition, it is important to highlight that prophylactic administration may cause antimicrobial resistance, and the long-term side effects may outweigh the beneficial effects.

Due to the significant variety in participant pools, severity of COPD, duration of treatment, intervention of choices, and outcomes measured between studies, the data were difficult to compare or generalize. Future studies need to focus on the minimum dosage of antibiotics required to achieve the desired effects and consider the application of prophylactic antibiotic medications during seasons associated with a high incidence of bacterial infections. Whether the efficacy of the macrolide prophylaxis in COPD patients can be attributed to antibacterial or immunomodulant properties, additional data are needed to reveal the exact drug mechanisms and interactions during exacerbations as well.

## Figures and Tables

**Figure 1 antibiotics-13-01110-f001:**
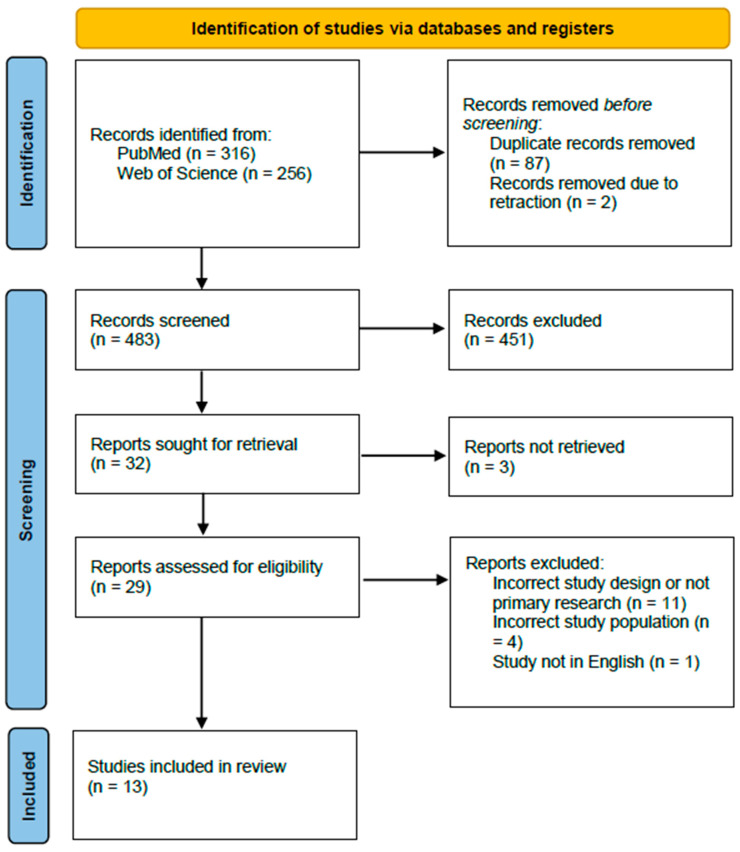
PRISMA flow diagram.

**Table 1 antibiotics-13-01110-t001:** Characteristics of the included studies.

Author/Year	Study Design	Intervention	Participants (Treatment/Control)	Result
Albert et al. [26]	Prospective, parallel-group, placebo-controlled trial	Azithromycin vs. placebo	558:559	Decreased significantly in exacerbation frequency and improved in quality of life.
Allinson et al. [27]	Double-blind, randomized, placebo-controlled trial	Doxycycline vs. placebo	110:112	No significant decrease in exacerbation rate.
Banerjee et al. [28]	Prospective, double-blind, randomized controlled trial	Clarithromycin vs. placebo	31:36	No improvement in health status, bacterial load, or exacerbation rate.
Berkhof et al. [29]	Single-center, parallel-group, randomized, double-blind, placebo-controlled trial	Azithromycin vs. placebo	42:42	Improvement in cough-specific health status.
Brill et al. [30]	Single-blind, randomized, placebo-controlled trial	Moxifloxacin vs. Doxycycline vs. Azithromycin vs. placebo	25:25:25:24	No change in airway bacterial load. Large increase in antibiotic resistance.
Han et al. [31]	RCT	Azithromycin vs. placebo	557:556	Effective in preventing AECOPD requiring both antibiotic and steroid treatment.
He et al. [32]	Double-blind, randomized, placebo-controlled study	Erythromycin vs. placebo	18:18	Significant reduction in airway inflammation and exacerbation rate.
Seemungal et al. [33]	Double-blind, randomized, placebo-controlled study	Erythromycin vs. placebo	53:56	Significant reduction in exacerbation rate.
Sethi et al. [34]	Multicenter, parallel-group, double-blind, randomized, placebo-controlled trial	Moxifloxacin vs. placebo	569:580	Reduction in odds of exacerbation.
Shafuddin et al. [35]	RCT	Roxithromycin/Doxycycline vs. Roxithromycin vs. placebo	101:97:94	No reduction in COPD exacerbation.
Simpson et al. [36]	RCT	Azithromycin vs. placebo	15:15	Reduction in severe exacerbation, sputum neutrophils, CXCL8 levels, and bacterial load.
Tan et al. [37]	RCT	Erythromycin (12 months) vs. Erythromycin (6 months) vs. placebo	18:18:18	Reduction in airway inflammation, improvement in exercise capacity.
Uzun et al. [38]	Single-center, double-blind, randomized, placebo-controlled trial	Azithromycin vs. placebo	47:45	Significant reduction in exacerbation rate compared to placebo.

**Table 2 antibiotics-13-01110-t002:** Intervention, duration, and outcome measures.

Author/Year	Intervention	Duration	Outcome Measure
Albert et al. [26]	Azithromycin 250 mg once daily vs. placebo	12 months	Time to first acute exacerbation; exacerbation rate; health status; nasopharyngeal colonization
Allinson et al. [27]	Doxycycline 100 mg once daily vs. placebo	12 months	Exacerbation rate; lung function; health status
Banerjee et al. [28]	Clarithromycin 500 mg once daily vs. placebo	3 months	Exacerbation rate; lung function; sputum bacterial load; health status; exercise capacity; CRP
Berkhof et al. [29]	Azithromycin 250 mg 3 times/week vs. placebo	3 months	Cough-specific health status; exacerbation rate; lung function
Brill et al. [30]	Pulsed moxifloxacin 400 mg daily for 5 days every 4 weeks vs. doxycycline 100 mg daily vs. azithromycin 250 mg 3 times/week vs. placebo	13 weeks	Sputum bacterial load; antibiotic resistance; health status; lung function and inflammatory markers
Han et al. [31]	Azithromycin 250 mg daily vs. placebo	12 months	AECOPD rate; health status
He et al. [32]	Erythromycin 125 mg 3 times/day vs. placebo	6 months	Lung function; health status; exacerbation; sputum assessment for cell counts and inflammatory markers
Seemungal et al. [33]	Erythromycin 250 mg twice/day vs. placebo	12 months	Exacerbation rate; exacerbation duration; inflammation at exacerbation; lung function
Sethi et al. [34]	Moxifloxacin 400 mg daily vs. placebo for 5 days.	Repeat every 8 weeks for a total of 6 courses	Exacerbation rate; health status; lung function; sputum bacterial load
Shafuddin et al. [35]	Roxithromycin 300 mg and Doxycycline 100 mg daily vs. roxithromycin 300 mg daily vs. placebo	3 months	Exacerbation rate during and post-treatment; lung function; health status
Simpson et al. [36]	Azithromycin 250 mg daily vs. placebo	3 months	Exacerbation rate; sputum assessment for cell counts; bacterial load and inflammatory markers
Tan et al. [37]	Erythromycin 125 mg 3 times/day vs. placebo	12-month group and 6-month group	Lung function; sputum inflammatory cells; exercise capacity
Uzun et al. [38]	Azithromycin 500 mg 3 times/week	12 months	Exacerbation; lung function; exercise capacity; health status; CRP; white blood cell count

**Table 3 antibiotics-13-01110-t003:** Exacerbation rate comparison between antibiotic and placebo groups.

Study	Intervention	Type of Association Measurement	Measure of Association (95% CI; *p* Value) Compared to Placebo
Albert et al. [26]	Azithromycin vs. placebo	Rate ratio	**0.83 (0.72 to 0.95; *p* = 0.01)**
Allinson et al. [27]	Doxycycline vs. placebo	Rate ratio	0.85 (0.67 to 1.07; *p* = 0.16)
Brill et al. [30]	Moxifloxacin vs. Doxycycline vs. Azithromycin vs. placebo	Relative risk	Moxifloxacin 1.38 (0.62 to 3.10; *p* = 0.43) **Doxycycline 2.07 (0.99 to 4.35; *p* = 0.05)** Azithromycin 0.72 (0.30 to 1.71; *p* = 0.45)
He et al. [32]	Erythromycin vs. placebo	Relative risk	**0.55 (0.31 to 0.97; *p* = 0.042)**
Seemungal et al. [33]	Erythromycin vs. placebo	Rate ratio	**0.64 (0.48 to 0.85; *p* = 0.003)**
Simpson et al. [36]	Azithromycin vs. placebo	Rate ratio	0.38 (0.14 to 1.05; *p* = 0.062)
Uzun et al. [38]	Azithromycin vs. placebo	Rate ratio	**0.58 (0.42 to 0.79; *p* = 0.001)**

Bold text indicates statistical significance.

**Table 4 antibiotics-13-01110-t004:** Mean difference in SGRQ total score compared to placebo.

Study	Intervention	Mean Difference in SGRQ Total Score (95% CI; *p*-Value) Compared to Placebo
Allinson et al. [27]	Doxycycline vs. placebo	**5.2 units (1.44 to 9.00; *p* = 0.007)**
Berkhof et al. [29]	Azithromycin vs. placebo	**−7.4 units (−12.5 to −2.5; *p* = 0.004)**
Brill et al. [30]	Moxifloxacin vs. Doxycycline vs. Azithromycin vs. placebo	Moxifloxacin: −1.88 units (−8.59 to 4.84; *p* = 0.59) Doxycycline: 1.02 units (−5.28 to 7.31; *p* = 0.75) Azithromycin: −2.29 units (−8.43 to 3.86; *p* = 0.47)
Uzun et al. [38]	Azithromycin vs. placebo	**−4.2 units (−8.3 to −0.1; *p* = 0.043) ***

Bold text indicates statistical significance; (*) indicates the change did not persist until the end of the treatment period.

## Data Availability

All data are provided within the manuscript.

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
