# Peer review of "Efficacy of Prophylactic Antibiotics in COPD: A Systematic Review"

_antibiotics, 2024, doi:10.3390/antibiotics13121110_

Round 1

Reviewer 1 Report

Comments and Suggestions for Authors

This manuscript provides a comprehensive systematic review on the prophylactic use of antibiotics in patients with chronic obstructive pulmonary disease (COPD), addressing their impact on reducing exacerbations and improving patients' quality of life. The structure of the article is clear, with well-organized sections on research background, methods, results, and discussion. However, there are areas that can be improved to enhance clarity and logical flow.

1.The study objectives in the abstract could be more specific, highlighting the key research questions. While the introduction offers a detailed overview of COPD and its pathophysiology, more comprehensive citations are recommended, particularly on controversial topics surrounding antibiotic use in COPD management. Including recent studies (e.g., from 2023) would enhance the relevance and authority of the article.

2.The results section could be more concise, directly highlighting the main conclusions, which will help readers quickly grasp the key points.

3.The specific research question of this review should be more clearly articulated, such as the long-term efficacy and safety concerns of antibiotics in reducing COPD exacerbations.

4.The inclusion and exclusion criteria should be more explicitly detailed. For example, explain why the 4-week no-exacerbation requirement was chosen, as this will help strengthen the methodological rigor.

5.The statistical analysis methods used for the results are not fully explained. It is recommended to include details about the statistical methods applied (e.g., p-value thresholds for statistical significance).

6.The discussion section could delve deeper into the immunomodulatory effects of antibiotics and the potential mechanisms behind antibiotic resistance. For the inconsistent results, suggestions for future research to resolve these discrepancies should also be elaborated.

7.The conclusions should be more definitive, offering direct recommendations for clinical practice regarding which types of antibiotics to use, and when they may not be appropriate.

Comments on the Quality of English Language

Some sentences are overly long and could be simplified. For example, “The uncontrolled activity of neutrophil elastase may result in the degradation of lung interstitial elastin” could be rephrased more concisely.

Author Response

Response to Comments:

Thank you for your time and consideration in reviewing our submission. We appreciate your valuable feedback and suggestions for improving the quality of our manuscript. Please, find the detailed responses below and the corresponding revisions highlighted in yellow in the re-submitted file.

Comment 1: This manuscript provides a comprehensive systematic review on the prophylactic use of antibiotics in patients with chronic obstructive pulmonary disease (COPD), addressing their impact on reducing exacerbations and improving patients' quality of life. The structure of the article is clear, with well-organized sections on research background, methods, results, and discussion. However, there are areas that can be improved to enhance clarity and logical flow.

The study objectives in the abstract could be more specific, highlighting the key research questions. While the introduction offers a detailed overview of COPD and its pathophysiology, more comprehensive citations are recommended, particularly on controversial topics surrounding antibiotic use in COPD management. Including recent studies (e.g., from 2023) would enhance the relevance and authority of the article.

Response 1: According to your recommendation, we have completed the abstract with the main key questions of our research e.g. exacerbation rate, time to first exacerbation, health status, airway bacterial load, inflammatory markers, cell counts in sputum samples and potential adverse events. Please, find the modifications on page 1, between lines 16-20. We appreciate your recommendation regarding the main concerns of antibiotic management of COPD. We find it to be valuable additions to our Introduction. Accordingly, we have incorporated relevant citations. Please, find the modifications highlighted on page 1, between lines 16-20 and on page 3, between lines 128-130 of our manuscript.

Comment 2: The results section could be more concise, directly highlighting the main conclusions, which will help readers quickly grasp the key points.

Response 2: In response to your feedback, we have updated and restructured the Results section to clarify findings, specifically the subchapter “Exacerbation” that we have separated to statistically significant and non-significant results. Please, find the updated section highlighted on page 8, between lines 200-219 of our revised manuscript.

Comment 3: The specific research question of this review should be more clearly articulated, such as the long-term efficacy and safety concerns of antibiotics in reducing COPD exacerbations.

Response 3: Thank you for your constructive suggestion. We have specified our research questions including the number of adverse events in the “Introduction” chapter on page 3 , between lines 130-136.

Comment 4: The inclusion and exclusion criteria should be more explicitly detailed. For example, explain why the 4-week no-exacerbation requirement was chosen, as this will help strengthen the methodological rigor.

Response 4: We appreciate your suggestion, we completed the “Study Participants” subchapter with the additional information and a citation regarding the 4-week exacerbation-free period to provide stronger support to our methodology. Please, find the modification on page 7, between lines 175-177.

Comment 5: The statistical analysis methods used for the results are not fully explained. It is recommended to include details about the statistical methods applied (e.g., p-value thresholds for statistical significance).

Response 5: Thank you for your suggestion, we have supplemented the “Methodology” part of the manuscript with the significance level of the statistical analyses used by the studies involved to facilitate the interpretation of our results. Please, find the modification on page 14 between lines 528-529.

Comment 6: The discussion section could delve deeper into the immunomodulatory effects of antibiotics and the potential mechanisms behind antibiotic resistance. For the inconsistent results, suggestions for future research to resolve these discrepancies should also be elaborated.

Response 6: According to your recommendations, we have incorporated potential mechanisms regarding antibiotic resistance and immunomodulatory effect with relevant citations to our “Discussion” chapter to make it more comprehensive. Please, find the modifications on page 12, between lines 416-425 and on page 13, between lines 439-454, respectively.

Based on the current findings, we suggest future studies to focus on the minimum dosage of antibiotics required to achieve the desired effects, and consider the application of prophylactic antibiotic medications during seasons associated with high incidence of bacterial infections. Please, find adjustments in our “Conclusions and Future Directions” chapter on page 15, between lines 550-553.

Comment 7: The conclusions should be more definitive, offering direct recommendations for clinical practice regarding which types of antibiotics to use, and when they may not be appropriate.

Response 7: We sincerely appreciate your valuable recommendation. Based on the included studies, we have expanded the “Conclusion” chapter with a paragraph with more specific recommendation, outlining the benefits and possible contraindications regarding azithromycin that may be useful in clinical practice. Please, find the supplementation on page 15, between lines 539-547.

Once again, we extend our sincere thanks and gratitude for your thorough review and insightful comments, which have been vital in increasing the quality and clarity of our manuscript. We are hopeful that the revisions and explanations provided address your concerns and contribute to the strength of our study. We remain available to answer any further questions you may have.

Sincerely,

Ágnes Tóth, PhD

Assistant Professor, Department of Integrative Health Sciences, Institute of Health Sciences, Faculty of Health Sciences, University of Debrecen

on behalf of all authors,

Anh Tuan Tran, Amr Sayed Ghanem, Marianna Móré and Attila Csaba Nagy

Reviewer 2 Report

Comments and Suggestions for Authors

The study is interesting and well designed and presented. Discussion is well developed and contain critical criteria to arrive to the conclusions. I only suggest to add some comments on what stated in lines 357, 360 and 418

Author Response

Response to Comments:

Thank you for your time and consideration in reviewing our submission. We sincerely appreciate your valuable and encouraging comments regarding our manuscript. Please, find the responses below and the corresponding revisions highlighted in blue in the re-submitted file.

Comment 1: The study is interesting and well designed and presented. Discussion is well developed and contain critical criteria to arrive to the conclusions. I only suggest to add some comments on what stated in lines 357, 360 and 418.

Response 1: According to your recommendations, we have incorporated additional comments into the “Discussion” chapter of our manuscript. Please, find the modifications on page 12, between lines 382-388 and 390-391 and on page 13, between lines 468-469.

Once again, we extend our sincere thanks and gratitude for your thorough review and insightful comments, which have been vital in increasing the quality and clarity of our manuscript. We are hopeful that the revisions and explanations provided address your concerns and contribute to the strength of our study. We remain available to answer any further questions you may have.

Sincerely,

Ágnes Tóth, PhD

Assistant Professor, Department of Integrative Health Sciences, Institute of Health Sciences, Faculty of Health Sciences, University of Debrecen

on behalf of all authors,

Anh Tuan Tran, Amr Sayed Ghanem, Marianna Móré and Attila Csaba Nagy

Reviewer 3 Report

Comments and Suggestions for Authors

General symptoms

Interesting study analyzing current evidence in RCTs in the literature exploring the effects of antibiotic prophylaxis in COPD.

The introduction is well descriptive, the methodology is sound but needs further clarifications, the results are clearly presented and detailed.

Please note that there are several grammatical and other language errors throughout the text which affect the quality of the manuscript.

Abstract

Abstract needs improvements in language quality and wording; for example: “detrimental effects” can be interpreted in many ways; “Prophylactic antibiotic approach utilizing macrolides is the most effective” – please clarify that this is based on current evidence from RCTs. Please also take care to correct language errors.

Introduction

Lines 55-80: please fragment.

Line 128 “This systematic review aims to summarize the efficacy of prophylactic antibiotic treatment for COPD exacerbations of previous studies” please rephrase.

Results

To better guide the reader, please provide a paragraph summarizing of your studies, including countries, total number of participants, demographics, which studies evaluated inflammatory markers, a summary of antibiotics used, etc.

Line 137: “11 reports were excluded due to wrong study design or not primary research, 4 reports were excluded due to wrong study population,” Please explain.

Table titles don’t need explanations, only a descriptive title. These explanations should be in the manuscript text.

Was staging of COPD correlated with study findings?

Methods

The methodology section needs expansion and clarifications. Please add study outcomes and conform to internationally standardized frameworks (eg.PRISMA).

In the Inclusion criteria, authors mention antibiotic treatment versus placebo. Please clarify that it was not treatment but prophylaxis.

Comments on the Quality of English Language

There are several grammatical and other language errors throughout the text which affect the quality of the manuscript.

Author Response

Response to Comments:

Thank you for your time and consideration in reviewing our submission. We appreciate your valuable feedback and suggestions for improving the quality of our manuscript. Please, find the responses below and the corresponding revisions highlighted in the re-submitted file.

Comment 1: General symptoms - Interesting study analyzing current evidence in RCTs in the literature exploring the effects of antibiotic prophylaxis in COPD.

The introduction is well descriptive, the methodology is sound but needs further clarifications, the results are clearly presented and detailed.

Please note that there are several grammatical and other language errors throughout the text which affect the quality of the manuscript.

Response 1:  Thank you for your constructive suggestions. Accordingly, we have enhanced our “Methodology” chapter. Please, find the modifications highlighted in green on pages 13-14, between lines 473-529. We agree with your comment regarding the English language, unfortunately there were several errors in the text. We have corrected them to improve the quality of our manuscript and better express our research. Please, find the English language-related modifications in pink.

Comment 2: Abstract - Abstract needs improvements in language quality and wording; for example: “detrimental effects” can be interpreted in many ways; “Prophylactic antibiotic approach utilizing macrolides is the most effective” – please clarify that this is based on current evidence from RCTs. Please also take care to correct language errors.

Response 2: Thank you for your suggestion that we need to note that the findings are based on current evidence from RCTs. We incorporated it into the “Abstract” between lines 28-29 as well as into the “Conclusions and Future Directions” chapter on page 15, in line 532. In response to your feedback, English has been improved in the “Abstract”. Please, find the corrections in pink. Instead of “detrimental effects” we used “negative effects” highlighted in pink on page 1, in line 24.

Comment 3: Introduction - Lines 55-80: please fragment.

Response 3: We appreciate your suggestion, accordingly, we have fragmented the text into two parts in the Introduction (page 2). The first paragraph (between lines 56-62) is focused on explaining the causes of COPD, the second paragraph (between lines 63-82) details the characteristic pattern of inflammatory responses in COPD patients.

Comment 4: Line 128 “This systematic review aims to summarize the efficacy of prophylactic antibiotic treatment for COPD exacerbations of previous studies” please rephrase.

Response 4: We have rephrased the sentence, i.e. “This systematic review aims to evaluate the effects of antibiotic prophylaxis on COPD patients’ exacerbations and quality of life, as well as safety concerns related to long-term administration of antibiotics.” Please, find the modification highlighted in green on page 3, between lines 130-132.

Comment: 5: Results - To better guide the reader, please provide a paragraph summarizing of your studies, including countries, total number of participants, demographics, which studies evaluated inflammatory markers, a summary of antibiotics used, etc.

Response 5: Thank you for your constructive suggestion. Accordingly, we have adjusted a summary subchapter of the relevant studies titled “Overview of Involved Studies” including the outcome measures and the antibiotic interventions, while the subsequent subchapter titled “Study Participants” had been supplemented with additional information about the number of participants and the including countries. Please, find the modifications on page 7, between lines 157-170 and on page 8, between lines 195-197, respectively.

Comment 6: Line 137: “11 reports were excluded due to wrong study design or not primary research, 4 reports were excluded due to wrong study population,” Please explain.

Response 6: Thank you for drawing our attention to the need for more specific explanations. In the wrong study design, the studies were mainly secondary, re-evaluation or retrospective observational studies. We considered patients with comorbidities (e.g. asthma, bronchiectasis, cystic fibrosis, antibody deficiency, tracheostomy etc.) or patients under the age of 18 as wrong study population. Please, find the adjusted clarifications in green on page 3, between lines 145-148.

Comment 7: Table titles don’t need explanations, only a descriptive title. These explanations should be in the manuscript text.

Response 7: Thank you for your comment, we have removed the explanations from the tables. Please, find the modifications on page 4, in line 153 (Table1), on page 6, in line 155 (Table 2), on page 8, in line 221 (Table 3) and on page 9, in line 262 (Table 4).

Comment 8: Was staging of COPD correlated with study findings?

Response 8: Thank you for your question. We currently do not have data on this aspect. Included studies did not classify the participants according to the stage of the disease. There was no measurement of correlation between COPD stage and the study finding about the antibiotic effects.

Comment 9: Methods - The methodology section needs expansion and clarifications. Please add study outcomes and conform to internationally standardized frameworks (eg.PRISMA).

In the Inclusion criteria, authors mention antibiotic treatment versus placebo. Please clarify that it was not treatment but prophylaxis.

Response 9: We appreciate your constructive recommendations. Therefore, we have developed and detailed the “Methodology” chapter, including “Type of Studies”, “Type of Participants”, “Type of Interventions”, “Type of Outcome Measures”, “Search Strategy”, “Selection of Studies” and “Data extraction” to provide a stronger support to our methods. Please, find the modifications on pages 13-14, between lines 473-529. Thank you for drawing our attention for avoiding the expression of “antibiotic treatment”. Instead, we used antibiotic “prophylaxis” “intervention” or “administration” in the whole text of the manuscript to prevent misunderstandings. Please, find the modifications in green on page 1, in line 35, on page 10, in lines 285, 298, 314, on page 10, in line 368, on page 12, in lines 410-411, 415 and on page 13, in line 462.

Comment 10: There are several grammatical and other language errors throughout the text which affect the quality of the manuscript.

Response 10: Thank you again for your notification about the grammatical and other language errors. We have revised the manuscript and we hope that we have managed to make the text clearer, grammatically more consistent and eliminate errors to improve the quality of our manuscript. Please, find the English language-related modifications in pink throughout the whole text.

Once again, we extend our sincere thanks and gratitude for your thorough review and insightful comments, which have been vital in increasing the quality and clarity of our manuscript. We are hopeful that the revisions and explanations provided address your concerns and contribute to the strength of our study. We remain available to answer any further questions you may have.

Sincerely,

Ágnes Tóth, PhD

Assistant Professor, Department of Integrative Health Sciences, Institute of Health Sciences, Faculty of Health Sciences, University of Debrecen

on behalf of all authors,

Anh Tuan Tran, Amr Sayed Ghanem, Marianna Móré and Attila Csaba Nagy

Round 2

Reviewer 1 Report

Comments and Suggestions for Authors

I think that the authors have adequately addressed my comments in the revised version of the manuscript.

Reviewer 3 Report

Comments and Suggestions for Authors

Thank you for the detailed response and amendments.